# Plasma Total Antioxidant Capacity and Carbonylated Proteins Are Increased in Pregnant Women with Severe COVID-19

**DOI:** 10.3390/v14040723

**Published:** 2022-03-30

**Authors:** Juan Mario Solis-Paredes, Araceli Montoya-Estrada, Adriana Cruz-Rico, Enrique Reyes-Muñoz, Javier Perez-Duran, Salvador Espino y Sosa, Victor Ranferi Garcia-Salgado, Rosalba Sevilla-Montoya, Raigam Jafet Martinez-Portilla, Guadalupe Estrada-Gutierrez, Juan Alexander Gomez-Ruiz, Paloma Mateu-Rogell, Jose Rafael Villafan-Bernal, Lourdes Rojas-Zepeda, Maria del Carmen Perez-Garcia, Johnatan Torres-Torres

**Affiliations:** 1Clinical Research Branch, Instituto Nacional de Perinatologia, Mexico City 11000, Mexico; juan.mario.sp@gmail.com (J.M.S.-P.); djavier40@gmail.com (J.P.-D.); salvadorespino@gmail.com (S.E.y.S.); vicranfery@gmail.com (V.R.G.-S.); rosalbasevilla@gmail.com (R.S.-M.); raifet@hotmail.com (R.J.M.-P.); gpestrad@gmail.com (G.E.-G.); dramateurogell@gmail.com (P.M.-R.); 2Coordination of Gynecological and Perinatal Endocrinology, Instituto Nacional de Perinatologia, Mexico City 11000, Mexico; ara_mones@hotmail.com (A.M.-E.); adriana.cruz@gmail.com (A.C.-R.); dr.enriquereyes@gmail.com (E.R.-M.); 3ABC Medical Center, Medical Association, Mexico City 05300, Mexico; 4Maternal-Fetal Medicine Department, Hospital General de Mexico “Dr. Eduardo Liceaga”, Mexico City 06720, Mexico; alexandergomezr4@gmail.com; 5Laboratory of Immunogenomics and Metabolic Diseases, Instituto Nacional de Medicina Genomica, Mexico City 14610, Mexico; joravibe@gmail.com; 6Maternal Fetal Medicine Department, Instituto Materno Infantil del Estado de México, Toluca 50170, Mexico; dra.rojaszepeda@gmail.com; 7Obstetrics & Gynecology Resident, Instituto Nacional de Perinatología, Mexico City 11000, Mexico; mcpg7@hotmail.com

**Keywords:** pregnancy, severe COVID-19, total antioxidant activity, carbonylated proteins

## Abstract

Oxidative stress (OS) induced by SARS-CoV-2 infection may play an important role in COVID-19 complications. However, information on oxidative damage in pregnant women with COVID-19 is limited. Objective: We aimed to compare lipid and protein oxidative damage and total antioxidant capacity (TAC) between pregnant women with severe and non-severe COVID-19. Methods: We studied a consecutive prospective cohort of patients admitted to the obstetrics emergency department. All women positive for SARS-CoV-2 infection by reverse transcription-polymerase chain reaction (RT-qPCR) were included. Clinical data were collected and blood samples were obtained at hospital admission. Plasma OS markers, malondialdehyde (MDA), carbonylated proteins (CP), and TAC; angiogenic markers, fms-like tyrosine kinase-1 (sFlt-1) and placental growth factor (PlGF); and renin-angiotensin system (RAS) markers, angiotensin-converting enzyme 2 (ACE-2) and angiotensin-II (ANG-II) were measured. Correlation between OS, angiogenic, and RAS was evaluated. Results: In total, 57 pregnant women with COVID-19 were included, 17 (28.9%) of which had severe COVID-19; there were 3 (5.30%) maternal deaths. Pregnant women with severe COVID-19 had higher levels of carbonylated proteins (5782 pmol vs. 6651 pmol; *p* = 0.024) and total antioxidant capacity (40.1 pmol vs. 56.1 pmol; *p* = 0.001) than women with non-severe COVID-19. TAC was negatively correlated with ANG-II (*p* < 0.0001) and MDA levels (*p* < 0.0001) and positively with the sFlt-1/PlGF ratio (*p* = 0.027). Conclusions: In pregnant women, severe COVID-19 is associated with an increase in protein oxidative damage and total antioxidant capacity as a possible counterregulatory mechanism.

## 1. Introduction

The pandemic caused by coronavirus 2 (SARS-CoV-2) and its symptomatic disease (COVID-19) has resulted in high morbidity and mortality [1], with pregnant women especially vulnerable to SARS-CoV-2 infection having even higher rates of mortality than the non-pregnant population [2]. SARS-CoV-2 infection can range from asymptomatic, to moderate, or life-threatening (severe) disease. Despite being known for its severe lung morbidity, SARS-CoV-2 infection may also cause damage to other organs, such as the brain, heart, blood vessels, liver, kidneys, and intestine [3]. Research studies have shown that SARS-CoV-2 disrupts the balance between pro-oxidant and antioxidant mediators, causing severe disease and lung injury [4], as observed in patients with cardiometabolic diseases, cancer, and chronic obstructive pulmonary diseases who display disturbed redox balance and have higher mortality rates and morbidity [5,6]. Oxidative stress (OS) plays a significant role in the pathogenesis of COVID-19, prolonging the cytokine storm cycle, promoting blood clotting, and exacerbating hypoxia. Furthermore, OS is associated with the generation of tissue lesions, mitochondrial dysfunction, and organ failure [7].

SARS-CoV-2 generates an imbalance in the expression of the NF-κB and Nrf2 pathways, two pathways responsible for maintaining the balance of cellular redox status and responses to stress and inflammation. Furthermore, it reduces the expression of angiotensin-converting enzyme 2 (ACE-2), resulting in the downregulation of angiotensin II type-2 receptor and placental growth factor (PlGF) and the upregulation of angiotensin II type-1 receptor and soluble fms-like tyrosine kinase-1 (sFlt-1) [8].

In pregnant women with preeclampsia, the overexpression of sFlt-1 promotes endothelial dysfunction and correlates with an increase in OS. Furthermore, high levels of sFlt-1 are related to the need for mechanical ventilation vasopressor support, favoring the development of severe acute kidney injury and death among critical patients [9,10]. We previously proposed that SARS-CoV-2 infection has an effect on the renin–angiotensin system (RAS) signaling pathway, causing an overproduction of reactive oxygen species (ROS), which could lead to oxidative stress generation, increasing sFlt1, and causing endothelial dysfunction, thus contributing to the pathogenesis of severe COVID-19 [11,12]. Considering the current evidence, we hypothesized that SARS-CoV-2 infection may increase OS in pregnant women with severe COVID-19.

Therefore, this research aims to evaluate the differences in total antioxidant capacity (TAC), oxidative damage markers (malondialdehyde [MDA] and carbonylated proteins [CP]), and angiogenic (sFlt-1, PlGF) and RAS (ANG-II, ACE-2) markers in pregnant women with severe and non-severe COVID-19.

## 2. Materials and Methods

### 2.1. Study Design and Participants

We conducted a cross-sectionally study at the National Institute of Perinatology—Isidro Espinosa de Los Reyes and the General Hospital de Mexico—Dr. Eduardo Liceaga in Mexico City. The inclusion criteria were all pregnant women who arrived at the emergency department with respiratory symptoms and a positive SARS-CoV-2 test by RT-qPCR between December 2020 and July 2021. The protocol was approved by the Institutional Review Board (IRB) of the National Institute of Perinatology (2020-1-32). All women provided written informed consent.

### 2.2. Data Collection and Plasma Measurement of OS, Angiogenic, and RAS Markers

Data on demographics and biochemicals characteristics were collected at hospital admission; an additional blood sample was obtained specifically for research purposes. The sample was centrifuged for 10 min at 1000× *g*. Plasma samples were aliquoted and stored at −70 °C until the assays were performed.

In the last stage of lipoperoxidation, when aliphatic aldehydes are formed, the most representative product, malondialdehyde (MDA), was quantified. MDA was measured using 1-methyl-2-phenylindole, by spectrophotometric assay at 586 nm [13]. Protein damage was evaluated by determining the exposure to carbonylated proteins (CP) in plasma following treatment with 2,4-dinitrophenylhydrazine, which reacts with carbonyl groups to form stable hydrazones. These were then measured spectrophotometrically at 370 nm according to the method described by Dalle Donne et al. [14], and expressed as pmol CP/mg protein. The TAC, indicative of the antioxidant defense system, was evaluated according to a method based on cupric-reducing antioxidant capacity (CUPRAC), using copper (II) and neocuproine reagents. The results were expressed as pmol Trolox equivalent/mg protein. Trolox is a water-soluble analog of vitamin E [15].

PlGF (Elecsys PlGF, Roche^®^, Basel, Switzerland) and sFlt-1 (Elecsys sFlt-1, Roche^®^) levels were measured by electrochemiluminescence using an automated analyzer Cobas-e411 (Roche Diagnostics^®^, Rotkreuz, Switzerland) according to the manufacturer’s instructions.

ELISA commercial kits were used to measure ACE-2 (Aviscera Bioscience, Santa Clara, CA, USA. cat SK00707-01) and ANG-II (Enzo Life Sciences, Farmingdale, NY, USA. cat ADI-900-204) according to the manufacturer’s instructions and analyzed in a Synergy HT plate reader (BioTek, Winooski, VT, USA).

### 2.3. Exposure

Included women were divided into exposed and non-exposed groups; exposure was considered as those who developed severe pneumonia. Severe pneumonia was defined according to the American Thoracic Society criteria [16,17].

Outcomes:

The main outcome was the TAC measured as pmol of Trolox equivalent/mg of protein, and the level of oxidative damage defined as the increase in malondialdehyde [MDA] and carbonylated proteins [CP] and angiogenic (sFlt-1, PlGF), and RAS (ANG-II, ACE-2) markers.

### 2.4. Statistical Analysis

Groups were divided according to exposure, namely severe or non-severe COVID-19. Data were analyzed using Student’s *t*-test to compare continuous variables with a normal distribution, whereas the Mann–Whitney U test was applied for comparisons of non-normally distributed variables. Spearman correlations for non-normally distributed samples were performed to study the correlation between angiogenic or RAS components and OS markers. Multivariate linear models were constructed to assess the association between exposures (severe vs. non-severe COVID-19) and the main outcomes (TAC and oxidative damage) adjusted for statistically significant confounders in the univariate analysis. Statistical significance was considered for *p* < 0.05. The program used was Prism 6.0 (GraphPad, San Diego, CA, USA).

## 3. Results

### 3.1. Characteristics of the Study Population

Of the 57 pregnant women with COVID-19 studied, 29.8% (*n* = 17) had severe COVID-19, including 3 (5.30%) maternal deaths. We found a difference in the median gestational age at hospital admission (34.6 in non-severe COVID-19 patients vs. 30.3 severe COVID-19 patients; *p* = 0.020) and in gestation age at delivery between groups. All deaths occurred in severe COVID-19 patients. The clinical characteristics of pregnant women according to exposure are shown in Table 1.

### 3.2. Association between OS, Angiogenic, and RAS Markers with the Primary and Secondary Outcomes

Pregnant women with severe COVID-19 had an increase of 15% in the levels of carbonylated proteins (5782 pmol vs. 6651 pmol; *p* = 0.024) and of 39.9% in the total antioxidant capacity (40.1 pmol vs. 56.1 pmol; *p* = 0.001), whereas no differences were observed in MDA levels (Figure 1).

In addition, in women with severe COVID-19, high levels of sFlt-1 (1477 vs. 3607; *p* = 0.033) were observed (Table 2).

Multilinear regression models were constructed to investigate associations between maternal characteristics and oxidative stress markers. COVID-19 was a binary variable (severe/non-severe), whereas maternal age, pregestational body mass index, and gestational age at diagnosis were continuous variables. The variable severe COVID-19 affected CP and TAC markers (Appendix A).

Spearman correlation between angiogenic or RAS components and OS markers revealed an inverse significant correlation of TAC with ANG II (*rho* = −0.594; *p* < 0.0001) and MDA levels (*rho* = −0.559; *p* < 0.0001). A positive significant correlation between TAC with sFlt-1/PlGF ratio (*rho* = 0.293; *p* = 0.027) was found.

## 4. Discussion

This study provides evidence that pregnant women with severe COVID-19 have increased in carbonylated proteins related to oxidative stress and an increase in total antioxidant capacity as a possible response mechanism.

SARS-CoV-2 infection progresses to the lower respiratory tract, particularly to alveolar epithelial cells, favoring the activation of alveolar macrophages and immune cells, generating a hypoxic and inflammatory environment leading to overproduction of ROS in alveoli and pulmonary microvessels [18], which contributes to the pathogenesis of viral infection [19]. In this study, we show for the first time that pregnant women with severe COVID-19 have higher levels of plasmatic CP. Information on protein damage in COVID-19 is limited; however, there is evidence that the oxidation of structural proteins occurs in the red blood cells of COVID-19 patients and may contribute to coagulopathic and thromboembolic events [20]. Protein oxidative damage is consistent with our previous hypothesis, where we proposed that SARS-CoV-2 pregnant women who developed severe COVID-19 have increased ROS levels [11].

The human body has mechanisms for the protection and prevention of oxidative damage, capable of defending cells against the toxic effects of free radicals [21] caused by several diseases such as diabetes mellitus, hypertension, coronary heart disease, and viral infections [22,23]. This mechanism is a measure of the total antioxidant capacity [21]. In our study, severe COVID-19 pregnant women had increase in TAC than the non-severe COVID-19 group. This TAC trend is similar to those reported in other populations with severe COVID-19 [22,24] and levels were found to be useful as a severity predictor [24]. Interestingly, TAC decreased in the group of patients who were admitted to the intensive unit care and required endotracheal intubation [22].

ROS molecules are products of cellular metabolism and involved in several pathways such as signal transduction, metabolism, cell differentiation, proliferation, and apoptosis. In addition, ROS participates in infection control as a protective mechanism for the host cell [25]. During the progression of viral replication, an excess of ROS is generated, leading to an imbalance in oxidation–reduction homeostasis, which could be involved in the modulation of cellular response, viral replication, host defense, and viral pathogenesis [26]. In this regard, the increase in TAC presented mainly by non-enzymatic antioxidants such as albumin, glutathione, ascorbic acid, a-tocopherol, and uric acid could be a response mechanism against oxidative damage caused by excess ROS [15,22,27]. However, it is not excluded that antioxidant activity of catalase, superoxide dismutase and glutathione-dependent enzymes in serum are also involved in oxidative damage control [28,29].

SARS-CoV-2 can infect endothelial cells, making COVID-19 a systemic vascular disease affecting endothelial function [30,31]. The vascular endothelium controls vascular relaxation and constriction, participates in fibrinolysis and coagulation, and regulates the immune responses [32].

Endothelial dysfunction has been reported in severe COVID-19; this condition is characterized by the reduction in vasodilators such as nitric oxide or increase in endothelium-derived contracting factors, leading to the impairment of endothelium-dependent vasodilation, tissue hypoxia, impaired perfusion, oxidative stress, inflammation, and subsequent organ failure [30,32].

Under normal circumstances, the peptidase activity of transmembrane protein ACE-2 cleaves angiotensin-II (ANG-II) into angiotensin 1-7 (Ang 1-7), conferring cardiovascular protection. It was previously reported that pregnant women with severe COVID-19 had lower ANG-II levels and we hypothesized that the spike protein of SARS-CoV-2 binds to trophoblastic cells expressing ACE-2, blocking the conversion of ANG-II into ANG 1-7 [11]. The accumulation of ANG-II on the cell surface enhances its binding to the AT-1 receptor (AT-1R), promoting downstream signaling, followed by rapid endocytosis of the ANG-II/AT-1R complex [33]. The excess of intracellular ANG-II in endothelial cells can bind to mitochondrial AT-1R, inducing cellular senescence with positive regulation of reactive oxygen species (ROS) [34,35,36]. Furthermore, the overactivation of AT-1R on the cell membrane leads to increased PKC, and calcineurin activity promotes nuclear factor-kappa B (NF-kB) and nuclear factor of activated T-cells (NFAT) pathways leading to an increase in gene expression and release of Flt-1. Flt-1 alternative splicing generates the sFlt-1 isoform [36]. The excess of sFlt-1 protein is released into the circulation causing endothelial dysfunction [11]. In our study, we observed a negative correlation between ANG-II and TAC, possibly as a response to oxidative insult, resulting in increased defense mechanisms according to the size of the aggression, neutralizing excess free radicals [21].

sFlt-1 is the soluble splice variant of the vascular endothelial growth factor receptor-1, which is present in the circulation and acts as an anti-angiogenic protein that antagonizes the vascular endothelial growth factor and PlGF [37]. Hypoxia is considered a trigger for release [38]. sFlt-1 is a key protein for endothelial dysfunction, and it is involved in generating a pro-oxidant environment [39]. Our study found that pregnant women with severe COVID-19 had higher levels of sFlt-1 than non-severe COVID-19. Interestingly, we found a positive correlation between sFlt-1/PlGF ratio and TAC, possibly as a protection mechanism in response to endothelial dysfunction and oxidative damage caused by sFlt-1 [12].

Finally, no differences were found in the levels of MDA between COVID-19 groups; this is similar to those reported in other COVID-19 populations not associated with pregnancy [40]. However, we found a negative correlation between TAC and MDA levels. Studies in cancer tissue found an increase in TAC and lower levels of MDA compared with adjacent normal tissue [41]; this effect could be similar to those observed in human viral infections where levels of antioxidant biomarkers increase to fight against oxidative compounds [22,26]. Therefore, we suggest that increased TAC could be a protective mechanism against lipid peroxidation caused by an excess of ROS, of which the final product is MDA [42].

### 4.1. Clinical Interpretation

Our results suggest that there may be an increase in oxidative stress markers in pregnant women with severe COVID-19 and that increased levels of TAC could act as a protective mechanism against oxidative damage and endothelial dysfunction in pregnant women with severe COVID-19. Although this is a cross-sectional study and no causal relationship can be made between severe COVID-19 and oxidative stress markers, the clinical implication of this work lies in the possible relationship between both and the development of antioxidant therapy, if proven in a cohort, to reduce the amount of oxidative stress that could be related to a severe spectrum of the disease.

### 4.2. Strengths and Limitations

The main strength of this study is the number of pregnant women with severe COVID-19, including three maternal deaths. These numbers are complicated to determine in developed countries and allow the comparison, in a more robust way, the extreme spectrums of the disease. In addition, very few articles evaluate oxidative markers combined with clinical information to allow statistical associations adjusted to clinical confounders for a better interpretation of the results.

However, despite these strengths, there are also limitations to this study. The most notable is the cross-sectional analytical design, as samples and clinical data were gathered at the time of hospital admission. This cross-sectional design does not allow us to make causal inferences between oxidative stress and severe COVID-19; thus, we cannot determine if, for women with chronic diseases such as diabetes, preeclampsia, obesity, the disease are the cause of the increase in oxidative stress and thus a cause for the development of severe disease, or if the established severity is the cause of increased oxidative stress. Further studies need to evaluate, in a prospective manner, the relationship between severity and oxidative stress to achieve a deeper understanding of the pathophysiology of the spectrum of severe COVID-19 in pregnant women.

## 5. Conclusions

Our findings suggest that there is a statistical association between severe COVID-19 and an increase in oxidative stress markers and total antioxidant capacity as a possible counter-regulatory mechanism in pregnant women with COVID-19. Further prospective research is needed to establish the causal relationship between oxidative stress and the spectrum of severe disease to allow the identification of possible knowledge gaps for future COVID-19 treatments.

## Figures and Tables

**Figure 1 viruses-14-00723-f001:**
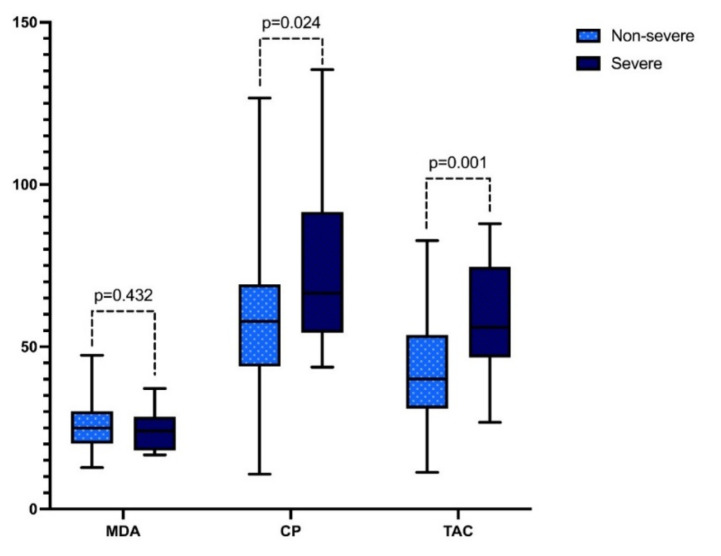
Oxidative stress markers in pregnant women with COVID-19. Malondialdehyde (MDA), carbonylated proteins (CP), and total antioxidant capacity (TAC).

**Table 1 viruses-14-00723-t001:** Clinical characteristics of COVID-19 pregnant women.

Characteristic	Non-Severe COVID-19 *n* = 40	Severe COVID-19 *n* = 17	*p*-Value
Maternal age (years)	30.50 (26.1–33.7)	31.5 (27.3–35.2)	0.458
Gestational age at diagnosis (weeks)	34.6 (31.0–38.6)	30.3 (27.1–33.6)	0.02
pBMI (kg/m^2^)	28.9 (24.9–31.2)	28.5 (23.3–31.6)	0.827
SpO2%	95.6 (93.0–96.0)	92.5 (79.6–96.0)	0.095
Smoking	0	1 (5.90%)	0.122
Chronic hypertension	1 (2.50%)	1 (5.90%)	0.495
Pre-gestational diabetes	2 (5.00%)	0	0.348
Chronic renal disease	1 (2.50%)	0	0.511
Gestational age at delivery (weeks)	38.1 (36.7–39.0)	35.0 (31.5–37.5)	0.002
Apgar 1 min	8 (7–8)	2 (2–8)	0.076
Apgar 5 min	9 (9–9)	9 (7–9)	0.126
Fetal growth restriction	3 (7.70%)	4 (25.0%)	0.08
Stillbirth	1 (2.60%)	1 (6.30%)	0.507
Neonatal death	2 (7.70%)	2 (16.7%)	0.402
ICU admission	4 (10.5%)	5 (29.4%)	0.08
Viral sepsis	7 (26.9%)	5 (41.7%)	0.363
Maternal death	0	3 (17.6%)	0.006

pBMI: pregestational body mass index; SpO2: Oxygen saturation; ICU: Intensive care unit. Mann–Whitney U test for continuous variables expressed as median and interquartile range; X^2^ or Fisher’s test for categorical variables expressed as number and percentage.

**Table 2 viruses-14-00723-t002:** Biochemical and hematological characteristics of pregnant women with COVID-19.

Characteristic	Non-Severe COVID-19	Severe COVID-19	*p*-Value
	*n* = 40	*n* = 17	
Leukocytes (×10/L)	8.65 (7.25–11.9)	8.40 (7.15–12.2)	0.884
Neutrophils (×10/L)	6.75 (5.25–9.60)	7.30 (4.95–10.6)	0.548
Lymphocytes (×10/L)	1.20 (0.90–1.63)	1.00 (0.85–1.20)	0.168
Hemoglobin (g/dL)	12.1 (11.3–13.0)	12.7 (11.2–14.2)	0.348
Hematocrit%	36.5 (33.9–39.3)	37.8 (34.7–42.0)	0.229
Platelets (×10^3^/L)	230 (200–275)	196 (174–238)	0.063
Glucose (mg/dL)	80.0 (73.0–86.5)	85.0 (75.5–102)	0.222
Triglycerides (mg/dL)	265 (204–320)	295 (214–345)	0.449
Total cholesterol (mg/dL)	197 (172–235)	143 (123–207)	0.015
Creatinine (mg/dl)	0.56 (0.56–0.68)	0.64 (0.53–0.75)	0.206
Uric acid (mg/dL)	4.30 (3.58–5.80)	4.40 (3.50–6.30)	0.951
D-Dimer (ng/mL)	2037 (1268–3598)	1434 (1276–2917)	0.58
Fibrinogen (mg/dL)	492 (446–603)	570 (456–615)	0.445
PTT (seconds)	27.0 (24.4–29.1)	26.2 (24.7–27.9)	0.774
PT (seconds)	11.0 (10.6–11.5)	10.2 (9.65–11.4)	0.011
C-RP (mg/L)	27.6 (9.64–106)	24.9 (9.86–129)	0.773
Procalcitonin (ng/mL)	0.07 (0.03–0.20)	0.25 (0.05–0.77)	0.038
MDA (pmol MDA/mg dry weight)	25.0 (20.1–30.1)	24.1 (18.0–28.5)	0.432
CP (pmol CP/mg of protein)	5782 (4393–6919)	6651 (5437–9154)	0.024
TAC (pmol of Trolox equivalent/mg of protein)	40.1 (31.0–53.6)	56.1 (46.8–74.7)	0.001
PlGF (pg/mL)	135 (48.7–195)	143 (52.4–281)	0.48
sFlt-1 (pg/mL)	1477 (1096–3275)	3607 (1936–8435)	0.033
sFlt1/PlGF ratio	18.0 (5.95–58.8)	30.3 (9.95–124)	0.383
ACE-2 (pg/mL)	7904 (6032–16400)	9480 (5621–28025)	0.789
ANG-II (pg/mL)	917 (595–2381)	549 (203–1073)	0.061

PTT: partial thromboplastin time; PT: prothrombin time; C-RP: C-reactive protein; MDA: malondialdehyde; CP: carbonylated proteins; TAC: total antioxidant capacity; PlGF: placental growth factor; sFlt-1: soluble fms-like tyrosine kinase-1; ACE-2: angiotensin-converting enzyme-2; ANG-II: angiotensin-II. Mann–Whitney U test for continuous variables expressed as median and interquartile range.

## Data Availability

The data presented in this study are available on request from the corresponding author. Data are not publicly available due to privacy.

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
