# Peer review of "Plasma Total Antioxidant Capacity and Carbonylated Proteins Are Increased in Pregnant Women with Severe COVID-19"

_viruses, 2022, doi:10.3390/v14040723_

Round 1
Reviewer 1 Report
The study aims to evaluate some peripheral markers of oxidative stress (OxS) in non-severe versus severe COVID-19 pregnant women. In the authors’ view the most relevant finding of their work was that total antioxidant capacity was higher in severe than non-severe COVID-19 pregnant women.
The manuscript is clear and well-written. However, the impact of the reported discoveries is very limited and result presentation as well as the study design suffer from severe shortcomings and flaws.
Major comments:
Abstract
- the type of test used for the statistical method should not be mentioned (especially when it is a very common one).
Methods:
- A control group of non-infected pregnant women should have been included
- None of the three markers used for detecting systemic OXS can be considered as accurate and/or reliable. MDA and carbonyls are not specific markers, in particular when they are measured by simple spectroscopic assays not preceded by HPLC and/or MS purification steps. 4-HNE and F2-isoprostanes (a assessed by MS) are much more suitable biomarkers of oxidative damage.
- The statistical analysis seems to referr to another study. Indeed, in this section, logistic regression and ROC curve were described, but there is no application of these tests in the analysis of the study results
Results
-multivariate analysis of the results was needed to check whether the difference observed between groups were independent of potential cofounders (age, comorbidities etc.)
Discussion
Due to the above mentioned weaknesses, the findings of the study does not significantly add to the current knowledge in the field.
The attempt of explaining the correlation between TAC and MDA is not convincing. The endogen contribution to TAC is marginal (the concentration of glutathione in blood is very low), thus it is not plausible that, quoting “High TAC could be a protective mechanism against lipid peroxidation caused by an ex-230 cess of ROS whose final product is MDA”. A possible balancing response to high OXS, could be due to antioxidant enzymes (SOD. CAT GPX), that, however, are much more active and effective into the cells than in plasma
Author Response
Consulte el archivo adjunto

Reviewer 2 Report
This is a very interesting study dealing with an important hot topic for the medical and scientific community.
Minor Point:
The authors have found increased oxidative damage and endothelial dysfunction in severe COVID-19 pregnant women. These results are quite important beyond the gestational age, and this requires an additional short comment in the discussion section, which would be very valuable for the readers. For instance, you may add in the Discussion section something like:
It is well know that patients with diabetes and obesity are among those more exposed to COVID-19 severity and deaths, although the mechanisms involved are not fully elucidated (Metab Syndr Relat Disord 2020;18:173-175). Findings from the present study suggest that there is a potential molecular explanation, since patients with diabetes and cardiometabolic diseases usually have increased oxidative damage and endothelial dysfunction (Transl Res 2009;153.217-223) and therefore, in case of impaired patients’ antioxidant capacity, this may explain why these categories of patients are more exposed to COVID-19 complications (Front Cardiovasc Med 2021;8:787761).
Reviewer 3 Report
In this manuscript, Solis-Paredes Juan Mario et al investigate the plasma lipid and protein oxidation expressed by malondialdehyde and carbonylated proteins levels and the total endogenous antioxidant status expressed by total antioxidant capacity (TAC) in COVID-19 pregnant women. The present manuscript highlights on one hand that in sever COVID-19 women the plasma carbonylated proteins level and TAC significantly increase and on the other hand TAC is negatively correlated with Angiotensin II and MDA, and positively correlated with sFlt-1 and sFlt-1/PIGF ratio, respectively.
The manuscript presents interesting results; however, some clearer specifications should be made for a better understanding of the study.
- The title does not appear to be the best version of this manuscript “Total antioxidant capacity and carbonylated proteins are increased in severe COVID-19 pregnant women”.
Only the nouns are frequently used in the title and not the sentences with verb.
Eg.
- The increase / the enhance of total antioxidant capacity and carbonylated proteins in severe COVID-19 pregnant women.
- The plasma oxidative stress status in the severe COVID-19 pregnant women etc.
However, the choice of the title is at the discretion of the authors.
- I propose you to use the expression lipid and protein oxidative damage or the oxidative damage of lipids and proteins instead of „oxidative lipid and protein damage” (Abstract, lines 27-28).
- Abstract, lines 28-29: I suggest you to mention at the end of the phrase the number of patients introduced in your study eg. (n=57) and if it is the case to rephrase or delete the next sentence.
- Abstract, line 32 – I suggest you to mention “plasma oxidative stress markers” because you assessed the oxidative stress markers from plasma and not from the tissues.
- Abstract, lines 41-42 – the conclusions of the abstract are very poor and NOT eloquent for the importance of conducting this study and for the results obtained.
I suggest you to rephrase the conclusions in both Abstract and the Conclusions section and highlight more results obtained and their importance.
- Keywords: the first keyword should be COVID-19 pregnant women or pregnancy but not severe COVID-19
- Line 52: “non pregnant population” → general population or non pregnant women
- Line 74: “…SARS-CoV-2 infection might alter OS status in severe COVID-19 pregnant women.”
I suppose that SARS-CoV-2 infection 1) might alter the general redox status or endogenous redox status or might alter the endogenous antioxidant status in severe COVID-19 pregnant women rather than OS status or 2) might enhance OS in severe COVID-19 pregnant women rather than OS status.
- Lines 74-77 should be replaced by the objective/ the aim of yours study. Why did you do this study? What you aimed to highlight with this study?
In the lines 66-72 you talk about SARS-CoV-2 and sFlt-1 and PIGF, respectively, and after that you say that in this study you determined the total antioxidant capacity, MDA and carbonyl group as a markers of lipid and protein oxidative damage. You don’t have a fluent transition from the information presented in introduction and the objective of the study, something is missing.
- Considering the same aspects in Introduction and in Discussion sections you should add more information about sFlt-1 and PIGT factors and their importance and relevance for SARS-CoV-2 infection or for your study (OS associated with SARS-CoV-2 infection).
In introduction (lines 66-69, 71-72) and Discussion (lines 218-222) you have mentioned something about sFlt-1 but I think that it is not enough.
Please also see the point 10 of the suggestion list.
- The lines 106-108:- Regarding the methods used to assess the markers of OS, I consider that you should mention some general aspects of each method used or the original method after which your protocol was adapted and only after that you can mention the bibliographic index.
Eg.
- Protein carbonyl content, an index of protein oxidative injury, was determined according to the spectrophotometric method described by Dalle Donne I et al (13).
- Malondialdehyde (MDA), the end-product of lipid peroxidation was assessed by a spectrophotometric method (586 nm) described by/adapted after Gérard-Monnier D et al (12).
- The lines 106-107 should be rephrase: “Oxidative lipid damage was assessed by quantification of MDA. A spectrophotometric assay was executed at 586 nm (12)”
Eg. Oxidative damage of lipids or lipid oxidative damage was assessed by quantification of malondialdehyde (MDA). MDA, the end-product of lipid peroxidation was assessed by a spectrophotometric method (586 nm) described by/adapted after Gérard-Monnier D et al (12).
- At statistical analysis (point 2.5, lines 136-143) you did not mention the p value for which you have considered the statistically significant differences between the groups compared, but then in the text you mentioned for some results the value of p.
- Lines 158-159: regarding the values of the parameters assessed, I suggest you to mention here the unit of measurement of these parameters and to delete these information (parameter values, unit of measurement) form the Legend of Figure 1. In Legend of the Figure you mention only what is represented in that Figure without the values and other information previously mentioned.
- I suggest you to split the lines 172-175 in 2 phrases to be more clear for the reader. Thus, the first sentence or phrase will present the inverse significant correlation between TAC and ANG II, TAC and MDA, and the second sentence will present the positive significant correlation between TAC and sFlt-1 (I suppose this) and also with the ratio sFlt-1/PIGF. In this variant the line 174-175 are not very clear, with whom sFlt-1 is positively correlated.
- Lines 177-178: “This study provides novel evidence that oxidative damage and TAC are altered in pregnant women with COVID-19.” I consider that the word “altered” is not appropriated in this context. Oxidative damage and TAC are both rather increased or enhanced in severe cases due the SARS-CoV-2 infection and the perturbation derived from it; The increase of TAC could be an adaptatif reaction of the body in the acute phase of a pathological process. After a period of oxidative damage, the plasma antioxidant system can be consumed and TAC values can decrease.
- Line 195: “This mechanism is known as total antioxidant capacity(25).” You can add here that the total antioxidant capacity are mainly represented by non-enzymatic antioxidants [Kohen et al] to be clear that the plasma antioxidant enzymes are not included.
Kohen, R.; Nyska, A. Oxidation of biological systems: Oxidative stress phenomena, antioxidants, redox reactions, and methods for their quantification. Toxicol. Pathol. 2002, 30, 620–650
- Lines 202-203: “ROS participate in infection control and are considerate a protective mechanism for the host cell.”
I suggest you to rephrase or even give more details regarding the roles of ROS.
In phagocytosis your statement may be true, but in another context not so much; otherwise the synthesis of ROS would be very beneficial. ROS are rather the intracellular second messengers that can be involved in the regulation of multiple signaling pathways.
- Line 212 “… SARS-CoV-2 bind to the ACE-2 receptors” – I consider that this formulation is not appropriate. SARS-CoV-2 uses the membrane ACE-2 as a mode of intracellular entry, so that ACE2 can be considered the SARS-CoV2 receptor. But there is not a receptor of ACE-2 that is used by SARS-CoV-2.
- In the line 211-216, I suppose that you directly link/connect ACE 2 with ANG II, but they are in inverse proportionality. Please play attention of two axis/branches of RAS system (ACE/Ang II/AT1 receptors and ACE2/Ang 1-7/Mas receptors) and you could even add some information regarding ACE2 and ANG II effects or implications in SARS-CoV-2 infection.
In the line (218-222) I suppose that you did not highlight enough the role/implication of sFlt-1 (see the point 10 of the suggestion list). A high level of sFlt-1 could confirm an endothelial dysfunction, pathophysiological disorder frequently reported in severe cases of SARS-CoV-2 infection.
Line 240-241 – For a while, TAC can be a good alternative of endogen antioxidants systems to protect the body from oxidative damage in SARS-CoV-2 infection.
In addition to the small or larger observations already made, I consider that the main weaknesses of this manuscript that I would recommend to be improved are:
- Discussion section where you did not discuss/detail enough your results in your context compared to what it was already published regarding SARS-CoV-2 infection or other critical pathologies. You should also better link the information together and make the switch from one aspect to another easier / more fluent: ROS - RAS - SARS-CoV2 infection and the parameters assessed in your study. In my opinion all these aspects could be linked much better.
Due to the fact that the novelties of your study could be the two factors (sFlt-1 and PIGF), I think you should insist a little bit more on them. TAC, carbonylated proteins, MDA, they are known and assessed in different pathologies and they are considered general markers of OS, not something very specific.
- As mentioned above, the Conclusions need to be improved and adapted to your results and assumptions/hypothesis.
Author Response
Consulte el archivo adjunto.

Round 2
Reviewer 3 Report
The authors answered the questions in a partially appropriate manner.
There are some small observations to make at the last manuscript.
- Line 70 -71 – I suggest you to mention the bibliographic index. I suppose that in the brackets you mentioned doi of the article used as bibliography.
- Line 205: Dysfunction endothelial → endothelial dysfunction is more appropriate.
- Lines 217-224; 249-251; 258-259 – please check the English and the punctuation used. There are a lot of omissions or mistakes, even missing punctuation marks.
Eg. ” sFlt-1 a key protein for endothelial dysfunction and generation of the pro-oxidant environ- 221 ment(41).” → sFlt-1 is a key protein for endothelial dysfunction and it is involved /responsible for generation of the pro-oxidant environment (41).
You can rephrase how you want but it is important to be clear for the readers and linguistically correct.
- The answer to question 20 is not clear in the manuscript. I did not see your answer.
I still have the idea that you do not make a clear difference between ANG II and ACE 2, that is, between the two branches of RAS. For a clear article that wants to be published in a well-rated journal (impact factor >5), I think it is important that all the aspects/issues are clear enough for the readers.
Author Response
Consulte el archivo adjunto.
